# Isokinetic Testing: Sensitivity of the Force-Velocity Relationship Assessed through the Two-Point Method to Discriminate between Muscle Groups and Participants’ Physical Activity Levels

**DOI:** 10.3390/ijerph17228570

**Published:** 2020-11-19

**Authors:** Danica Janicijevic, Olivera M. Knezevic, Amador Garcia-Ramos, Danilo Cvetic, Dragan M. Mirkov

**Affiliations:** 1Faculty of Sport and Physical Education, University of Belgrade, 11030 Belgrade, Serbia; jan.danica@gmail.com (D.J.); olivera.knezevic@imi.bg.ac.rs (O.M.K.); cveticdanilo@yahoo.com (D.C.); dragan.mirkov@fsfv.bg.ac.rs (D.M.M.); 2Institute for Medical Research, University of Belgrade, 11129 Belgrade, Serbia; 3Department of Physical Education and Sport, Faculty of Sport Sciences, University of Granada, 18011 Granada, Spain; 4Department of Sports Sciences and Physical Conditioning, Faculty of Education, Universidad Católica de la Santísima Concepción, Concepción 4030000, Chile

**Keywords:** flexors, extensors, knee, hip, elbow, shoulder

## Abstract

Background: Isokinetic testing has been routinely used to assess the capacities of individual muscle groups. In this study we aimed to evaluate the sensitivity of the force-point (F-v) relationship assessed through the two-point method to discriminate between antagonist muscle groups and males with different physical activity levels. Methods: The concentric force output of the knee, hip, elbow, and shoulder flexors and extensors of 27 active and 13 non-active men was recorded at 60 and 180°/s to determine the F-v relationship parameters (maximum force [*F*_0_], maximum velocity [*v*_0_], and maximum power [P_max_]). Results: *F*_0_ and P_max_ were higher for knee extensors (effect size [ES] = 1.97 and 0.57, respectively), hip extensors (ES = 2.52 and 0.77, respectively), and shoulder flexors (ES = 1.67 and 0.83, respectively) compared to their antagonist muscles, while *v*_0_ was higher for knee flexors compared to knee extensors (ES = 0.59). Active males revealed higher *F*_0_ for knee extensors (ES = 0.72) and knee flexors (ES = 0.83) and higher P_max_ for knee flexors (ES = 0.70), elbow extensors (ES = 0.83) and shoulder extensors (ES = 0.36). Conclusions: The sensitivity of the two-point method for testing the maximal mechanical capacities was high for the knee, moderate for the hip and shoulder, and low for the elbow joint.

## 1. Introduction

Muscle isokinetic strength tests are considered safe, valid and reliable [1,2,3,4]. Therefore, they have been widely used to assess the maximal capacities of individual muscle groups to produce force and power as well as the balance ratios in these maximal capacities between the individual muscles [5,6]. A basic requisite of isokinetic testing is to record force output at a constant movement velocity, that may range from 0–500°/s depending on the device [7]. However, angular velocities above 180°/s have been discouraged when assessing isokinetic strength during concentric actions because the range of motion under a constant velocity is very small and this may compromise the accuracy of the measurement [8]. The two velocities most commonly used during isokinetic testing procedures are 60 and 180°/s [9,10,11], which have been suggested to reveal the maximal capacities of the muscles to produce force and power, respectively [12,13]. However, it is known that higher force outputs can be achieved under isokinetic tests performed at lower angular velocities, while maximal power could be attained under isokinetic test performed at higher angular velocities (i.e., >180°/s) [13]. In addition, the standard isokinetic test (i.e., force output recorded against a predetermined velocity) cannot reveal the maximal velocity capacity because (I) the velocity cannot be voluntarily changed during the movement, and (II) the maximal movement velocity is considerably higher than the velocities typically used during isokinetic tests [14].

Linear regression has been recommended for modelling the force-velocity (F-v) relationships during multi-joint movements since there is a strong evidence that the F-v relationship during these tasks follows a linear shape [15,16,17,18,19]. A direct consequence of the high linearity of the F-v relationship is the hyperbolic shape of the power-velocity relationship [15,16,17,18,19]. However, recent studies have suggested that the F-v relationship may also be linear when obtained from single-joint isokinetic tasks [9,10,20]. A benefit of the strong linearity of the F-v relationship is that it provides a possibility to estimate maximum force (*F*_0_), velocity (*v*_0_) and power (P_max_) capacities within a single testing procedure. In this manner, additional tests for separate evaluation of the *v*_0_ and P_max_ capacities could be avoided. Furthermore, recording force values against only two angular velocities could provide enough information to accurately determine the F-v relationship (i.e., two-point method) [11,17,21,22]. In this regard, Grbic et al. [10] reported a high validity of the two-point method (force recorded at 60 and 180°/s) compared to the multiple-point method (force recorded at five velocities: 30, 60, 120, 180, 240°/s) for exploring mechanical capacities of the knee extensors, while Janicijevic et al. [11] obtained high validity of the two-point method to estimate *F*_0_ of the knee extensors, knee flexors, elbow extensors and elbow flexors. However, to date, no study has evaluated the feasibility of the two-point method in isokinetic settings (e.g., 60 and 180°/s).

It would be important to elucidate whether the F-v relationship assessed through the two-point method is able to discriminate between participants of different physical activity levels (active vs. non-active) as well as between antagonistic muscle groups (e.g., knee, hip, elbow and shoulder). The F-v relationship parameters were effective to discriminate between jumping abilities of participants with different physical activity levels [16], however, to our knowledge, no previous study has used isokinetic dynamometry to explore whether the F-v relationship parameters obtained during single-joint movements differ between participants with different physical activity levels. In addition, the isokinetic dynamometry has been widely used to explore the strength balance ratios of the antagonist muscle groups, since it presents valuable additional information to the strength of individual muscles [23]. In this regard, higher forces have been reported for extensor muscles acting on the knee [24] and hip [25], while flexor muscles were stronger at shoulder [26] and elbow joints [27]. However, no previous study has used the two-point method to explore differences in the mechanical capacities of antagonistic muscle groups. Therefore, it could be interesting to explore if the F-v relationship modelling can help us discriminate not only between maximal force capacities (*F*_0_), but also between maximal power (P_max_) and maximal velocity capacities (*v*_0_) of antagonist muscle pairs.

Scarce information exists regarding the maximal velocity capacity of individual muscles groups. Findings from studies using other technologies (high speed cameras, goniometers, accelerometer etc.) for measuring the maximal velocity of different body segments [28,29,30,31] are quite unequivocal. For instance, Jessop and Pain [30] revealed that knee extensors, elbow extensors, shoulder extensors and hip extensors can achieve higher velocities than their antagonistic muscle pairs, while Jaric [31] found that elbow flexors can be shortened faster than elbow extensors in a variety of conditions, which was in line with the study of Mirkov et al. [32]. The issue with comparing maximal velocities between different studies was highlighted by Bober et al. [14] who reported that the maximal velocity of knee extensor muscles was dependent of both the range of motion and pre-stretch (velocity values ranged from 213 to 1087°/s). Therefore, the existing literature does not allow us to hypothesize regarding the possible differences in maximal velocity capacities between antagonist muscle pairs acting on several joints. Finally, it would be also interesting to determine the possibility of generalizing the outcomes of the F-v relationship between antagonist muscle pairs, since it could shorten the testing procedure (e.g., possible high correlations would allow the prediction of the mechanical capacities of one muscle group based on the results obtained in the antagonistic muscle group). High generalizability of the F-v parameters between antagonistic muscle groups could further motivate using the F-v relationship in routine isokinetic testing. Note that the generalizability of the F-v relationship parameters has been shown to be low between different muscle groups assessed during multi-joint task (e.g., jumping and sprinting) [33], but no previous study has explored the association between the same F-v relationship parameters obtained from antagonist muscle pairs assessed using isokinetic dynamometry.

To address the aforementioned issues, the main aim of this study was to evaluate whether the magnitude of the F-v relationship parameters (i.e., *F*_0_, *v*_0_, P_max_, and F-v slope) obtained using the two-point method (i.e., force output recorded against only two angular velocities) are sensitive to discriminate between flexors and extensors acting on the same joints (knee, hip, elbow and shoulder), and between men with different levels of physical activity (active vs. non-active). The generalizability of the same F-v relationship parameters between antagonist muscle pairs was also examined. We hypothesized that (I) *F*_0_ and P_max_ obtained during knee, hip, shoulder extension, and elbow flexion would be higher than during the knee, hip, shoulder flexion, and elbow extension, respectively. We also hypothesized that (II) *F*_0_ and P_max_ of all muscles would be higher for active compared to non-active subjects, and (III) the association between the same F-v relationship parameters across different antagonistic muscle groups will be low. These results could contribute to the better understanding of the isokinetic testing procedure of the F-v relationship through the two-point method.

## 2. Materials and Methods

### 2.1. Participants

A priori analysis (G*Power 3.1.9.4, Kiel University, Kiel, Germany) revealed that for performing the statistical analysis with a power of 0.95 and α = 0.05 the total sample size should consider 36 participants and, therefore, we conservatively recruited forty young men to participate in this study. The physical activity level was assessed by the International Physical Activity Questionnaire (IPAQ), which was used to divide participants in active (n = 27, age = 23.7 ± 2.9 years [range = 21.0–26.0 years], height = 1.83 ± 0.06 m, body mass: 79.8 ± 8.0 kg) and non-active group (n = 13, age = 21.9 ± 4.0 years [range = 17.8–26.0 years], height = 1.80 ± 0.06 m, body mass = 68.4 ± 9.9 kg). Participants were considered physically active if they were performing at least 5 h of moderate to highly intensive physical activity per week, while non-active participants reported complete absence of physical exercise. All participants were free from chronic diseases and musculoskeletal injuries. Participants were introduced with the testing procedures and possible risks associated with isokinetic assessment. The study protocol was approved by the Faculty of sport and physical education University of Belgrade Review Board on 31 May 2017 (Approval number: 02-856-2) and an informed consent in accordance with the principles of the Declaration of Helsinki was signed by all participants.

### 2.2. Study Design

A crossover study design was used to explore the feasibility of the two-point method for assessing the mechanical capacities of flexor and extensor muscle groups acting in various joints. The study consisted of four testing sessions separated by 48–72 h. The flexor and extensor muscle groups of one joint were tested in each session against two angular velocities. The order of testing of the joints (knee, hip, elbow and shoulder), muscles (flexors and extensors), and velocities (60 and 180°/s) was randomised. All sessions were performed at the same time of the day for each participant (±1 h) and under similar environmental conditions (~22 °C and ~60% humidity).

### 2.3. Testing Procedures

All measurements were conducted at the Faculty research laboratory, using an isokinetic dynamometer (Kin-Kom AP125, Chatex Corp., Chattanooga, TN, USA). Each testing session began with a standardised warm-up consisting of 5 min of cycling on a leg cycle ergometer and stretching exercises [34]. Afterwards, the participants were seated into the chair of the dynamometer and fixed with Velcro straps in accordance with the manufacturer’s guidelines. The axis of the joints was aligned with the axis of the dynamometer using visual inspection and manual palpation. Muscle force was assessed at two angular velocities: 60 and 180°/s. Participants performed three cycles of maximal voluntary concentric contractions (1 cycle = 1 flexion + 1 extension) separated by 30 s. In total, participants performed three trials of three cycles under each testing velocity, where the first trial was used for familiarization, and other two trials were used for statistical analyses. The recovery time between different sets was set to 2 min. Participants were encouraged by the same experienced examiner to perform the movement as fast and as hard as possible. In addition, participants received visual feedback of force values throughout the whole execution of the exercise. All measurements were performed with the dominant extremity (i.e., the one they would use for kicking a ball [knee and hip exercises] and writing [elbow and shoulder exercises]) [5,35]. The range of motion was 80° for the knee tasks (from 90° to 170°, 180° representing full extension) [10], 50° for the hip tasks (from 90° to 140°) [36], 65° for the elbow tasks (from 45° to 110°) [37], and 80° for the shoulder tasks (from 90° to 170°) [36].

### 2.4. Data Acquisition and Analysis

A custom-made LabView (National Instruments Corporation, Austin, TX, USA) application was used to provide visual feedback on a computer screen, data acquisition, and processing of the force-time signals. Force-time signals were recorded at 500 Hz and low-pass filtered (5 Hz) using a second-order (zero-phase lag) Butterworth filter. The peak force value of each trial was obtained from the isokinetic part of the force-time curve [8]. The highest peak force of the three trials was used for further analyses. Force data were normalized to the body mass on the power of 2/3 [38]. Linear velocities (m·s^−1^) were calculated by multiplying the angular velocity by the length of individuals’ lever arm. Then, using normalised force and linear velocity, F-v relationships were derived by fitting the following linear regression model:F(V) = *F*_0_ − aV(1)
where *F*_0_ represents the force-intercept (i.e., theoretical maximal force), a is the slope that corresponds to *F*_0_/*v*_0_, and *v*_0_ is the velocity-intercept (i.e., theoretical maximal velocity). As a direct consequence of the F-v relationship linearity, P_max_ was calculated as P_max_ = *F*_0_·*v*_0_/4. Gravity correction was performed by placing the lever arm as close as possible to horizontal position, but avoiding putting muscle antagonistic pairs in stretched positions. Therefore, gravity correction angle was 170° for knee tasks, 140° for hip tasks, and 90° for the elbow and shoulder tasks [39].

### 2.5. Statistical Analysis

The normal distribution of the data (Shapiro-Wilk test) and the homogeneity of variances (Levene’s test) were confirmed (*p* > 0.05). Descriptive data are presented as mean and standard deviation (SD), while the Pearson’s correlation coefficients (*r*) are presented through their median and inter-quartile range values. A total of 16 mixed-model ANOVAs with Bonferroni post hoc corrections (4 F-v relationship parameters [*F*_0_, *v*_0_, F-v slope and P_max_] × 4 joints [knee, hip, elbow and shoulder]) were applied with the muscle group (flexor vs. extensor) as within- and physical activity level (active vs. non-active) as between-participant factors. The Cohen’s d effect size (ES) was used to explore the magnitude of the differences and it was computed considering the harmonic mean of the SD of the compared conditions. The following scale was used to interpret the magnitude of the ES: negligible (<0.2), small (0.2–0.5), moderate (0.5–0.8), and large (≥0.8) [40]. The *r* coefficients were used to quantify the magnitude of the associations between the same F-v relationship parameters obtained in the antagonistic muscle groups acting on the same joint. Qualitative interpretations of the *r* coefficients as defined by Hopkins et al. [41] (0.00–0.09 trivial; 0.10–0.29 small; 0.30–0.49 moderate; 0.50–0.69 large; 0.70–0.89 very large; 0.90–0.99 nearly perfect; 1.00 perfect) were provided for all significant correlations. Magnitude-based inference was performed by means of a custom Excel spreadsheet, while other statistical analyses were performed using the software package SPSS (IBM SPSS version 22.0, Chicago, IL, USA). Statistical significance was set at an alpha level of 0.05.

## 3. Results

None of the muscle group × physical activity level interactions reached statistical significance (*p* ≥ 0.093) (Table 1, Figure 1). A significant main effect of muscle group was observed for *F*_0_, F-v slope and P_max_ in the knee and hip joints (higher values for extensors) as well as in the shoulder joint (higher values for flexors), while for *v*_0_ the main effect of muscle group reached statistical significance only for the knee joint (higher value for flexors) (Figure 2). A significant main effect of physical activity level was found for *F*_0_ during the knee extension and knee flexion tasks and for P_max_ during knee flexion, elbow and shoulder extension tasks (higher values were always obtained by the active group) (Figure 3).

The correlations of the F-v relationship parameters between flexor and extensor muscles acting on the same joint are presented in Table 2. Moderate to large correlations were observed in the knee and shoulder joints for all F-v relationship parameters (r range from 0.349 to 0.571). Hip joint showed significant correlations for *F*_0_ (r = 0.640) and F-v slope (r = 0.385), while the elbow joint only showed a significant correlation for *F*_0_ (r = 0.513).

## 4. Discussion

This study was designed to explore whether the F-v relationship modelled by the two-point method could discriminate between antagonist muscle groups, and males with different physical activity levels. The main findings revealed that (I) *F*_0_, F-v slope and P_max_ were higher for the knee extensors, hip extensors and shoulder flexors compared to their antagonistic muscle pairs (knee flexors, hip flexors, shoulder extensors), while *v*_0_ was significantly higher for knee flexors compared to knee extensors, (II) *F*_0_ was higher for active compared to non-active males only during the knee extension and knee flexion tasks, while P_max_ was higher for active males during knee flexion, elbow extension and shoulder extension tasks, and (III) the association between the same F-v parameters across different muscle groups were generally moderate to large. The first two findings generally support the two-point method as a sensitive procedure for testing muscle capacities during knee, hip and shoulder isokinetic tasks, while a lower sensitivity was observed for the elbow task. The third finding suggests that the association of the F-v relationship parameters between antagonist muscle groups could be higher than the previously reported between different multi-joint exercises.

The function of the muscles acting on the knee joint has been commonly evaluated by isokinetic dynamometry [10]. Previous studies have reported higher values of force under isokinetic conditions for knee extensors compared to knee flexors [5,42]. Similarly, we observed higher values of *F*_0_ and P_max_ for the knee extensors compared to the knee flexors. Even though a specific hypothesis regarding *v*_0_ was not formulated, our results demonstrated that knee flexors tends to show a higher *v*_0_ than knee extensors. A plausible explanation might be the different architecture of knee extensors and knee flexors (i.e., knee flexors consist of muscle fibers that are positioned parallelly, while knee extensors consist from the fibers that present a greater pennation angle). In addition, both *F*_0_ and P_max_ were higher for active males during the knee flexion task, while only *F*_0_ was higher for active males during the knee extension task. Therefore, as far as the knee joint is concerned, it can be concluded that the two-point method was sensitive enough to discriminate between antagonist muscles as well as between males of different physical activity levels.

The weakness of the muscles acting on the hip joint may cause dynamic imbalance of the entire kinetic chain of the lower limbs [43]. In line with other studies [44,45], we observed both higher *F*_0_ and P_max_ for hip extensors compared to hip flexors, while no significant differences were observed between active and non-active males for any of the F-v relationship parameter. Therefore, while the two-point method seems to be effective to discriminate between hip extensors and hip flexors, it remains unclear whether it could also be sensitive to discriminate between males of different physical activity levels. Future studies should compare populations with clear differences in the strength of hip extensors and hip flexors (e.g., runners vs. taekwondo athletes) to further explore the sensitivity of the two-point method to discriminate between different populations.

The repetitive overhead movements which are common for some sports (e.g., throwing and spiking) can reach up to 2300°/s during overhead pitching [46] and 1700°/s for tennis serve [47]. These extremely high velocities emphasize the importance of developing the strength of the muscles acting on the elbow. Rejecting our hypothesis, no significant differences were found for any F-v relationship parameter between the elbow flexors and elbow extensors. This contradicts the results of Mirkov et al. [32] and Jaric [31] who revealed higher velocities for elbow flexors compared to elbow extensors under a variety of conditions. In addition, only P_max_ during the elbow extension task was significantly higher for active compared to non-active males. The overall lack of differences between active and non-active males could be explained by the fact that they did not necessarily differ in the activities performed with the upper limbs, or by the fact that isokinetic testing may not be sensitive to discriminate between active and non-active populations [48]. Future studies should explore whether the two-point method could be able to detect differences in the F-v relationship parameters between groups that clearly differ in the strength capacity of their elbow muscles. Therefore, based in our findings, the sensitivity of the two-point method for assessing the mechanical capacities of elbow flexors and elbow extensors should be elaborated by further studies.

Shoulder isokinetic testing is commonly used not only for testing subjects who are recovering from shoulder injuries, but also for healthy overhead athletes (i.e., those who use their upper limbs in an arc over head to propel a ball) [49]. The shoulder joint is one of the most mobile joints of the human body and, therefore, it needs to be surrounded with strong muscles [31]. Confirming our first hypothesis, higher *F*_0_ values were obtained for shoulder flexors compared to shoulder extensors. However, only P_max_ during shoulder extension task revealed higher values for active than for non-active males. Previous studies have reported higher force values for active subjects compared to non-active subjects during both shoulder extension and shoulder flexion tasks [26]. The absence of differences in *F*_0_ in our study could be explained because the level of upper limb activity did not meaningfully differ between the active and non-active groups. Note that the findings related to the shoulder joint are somehow similar to the ones reported for hip muscles, suggesting that the two-point method is sensitive to discriminate between flexor and extensor muscles, but a lower sensitivity was observed for discriminating between active and non-active males. Therefore, the recommendation of comparing groups with clear differences in upper-body force capacities could be also applied for the shoulder joint.

The possibility of generalising the F-v relationship parameters between antagonist muscle groups was also explored in the present study. Although rejecting our last hypothesis, we observed stronger correlations for the magnitude of the same F-v parameters than previous studies that explored the correlations between different isoinertial multi-joint exercises [50,51]. Regardless of the higher generalizability of the F-v relationship parameters observed in the present study, it should be noted that significant correlations were not systematically reached, which suggest that a given maximal mechanical capacity cannot be predicted from the value observed in the antagonist muscle group. Regarding the possible limitations of the current study, it should be acknowledged that during the testing procedure we applied the two most commonly used angular velocities (i.e., 60°/s and 180°/s). Because these two velocities are far from the velocity-intercept, it is possible that the precision of the F-v relationship could be improved using velocities closer to the velocity-intercept by reducing the extrapolation needed to reach *v*_0_ [17]. Therefore, it would be interesting to explore which combination of two velocities increases the accuracy in the determination of the F-v relationship parameters. It is plausible that more extreme angular velocities could be recommended for task with longer range of motion, but this hypothesis should be confirmed by future studies. In addition, the lack of significant differences between active and non-active males for several F-v relationship parameters and muscle groups could be the consequence of not controlling the type of sport and recreational activity performed by the subjects. Therefore, future studies should try to compare subjects with more distinctive characteristics regarding the function of the different muscles assessed.

## 5. Conclusions

The sensitivity of the two-point method for testing the maximal mechanical capacities was high for the knee joint, moderate for the hip and shoulder joints, and low for the elbow joint. The F-v relationship assessed through the two-point method was able to discriminate better between antagonist muscle groups than between males with different levels of physical activity. The non-systematic correlations between the F-v relationship parameters of antagonist muscle groups suggest that a given maximal mechanical capacity cannot be predicted from the value observed in the antagonist muscle group. Therefore, since different muscle groups should be evaluated to obtain comprehensive information of the function of the whole neuromuscular system, the two-point method could be considered as a quick procedure for testing the maximal mechanical capacities to produce force, velocity, and power.

## Figures and Tables

**Figure 1 ijerph-17-08570-f001:**
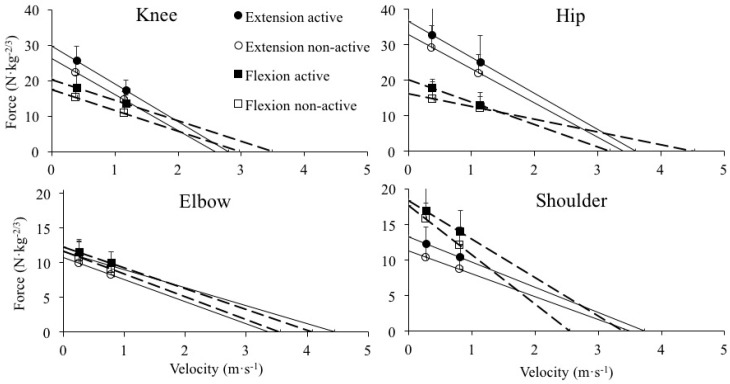
Linear regression models obtained from the force and velocity data averaged across the participants during the knee extension and flexion (**upper-left panel**), elbow extension and flexion (**lower-left panel**), hip extension and flexion (**upper-right panel**) and shoulder extension and flexion (**lower-right panel**) tasks. Straight and dashed lines represent extensor and flexor muscles, respectively. The error bars represent the standard deviation obtained by flexors (squares) and extensors (circles) groups.

**Figure 2 ijerph-17-08570-f002:**
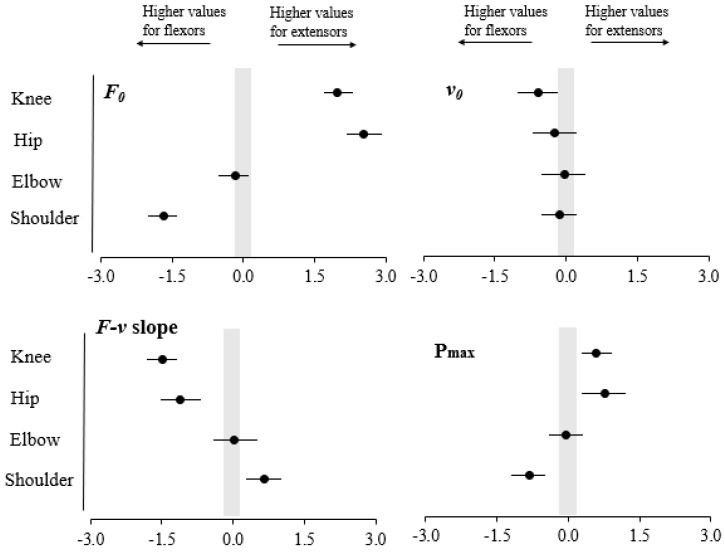
Standardized differences (Effect sizes and 95% confidence intervals) for maximum force (*F*_0_; **upper-left panel**), maximum velocity (*v*_0_; **upper-right panel**), force-velocity slope (F-v slope; **lower-left panel**) and maximum power (P_max_; **lower-right panel**) between the antagonist muscle pairs acting on the knee, hip, elbow and shoulder joints (Effect size = Extensor mean − Flexor mean/SDboth). The probability that the true difference was trivial (ES from −0.20 to 0.20) or substantial is depicted.

**Figure 3 ijerph-17-08570-f003:**
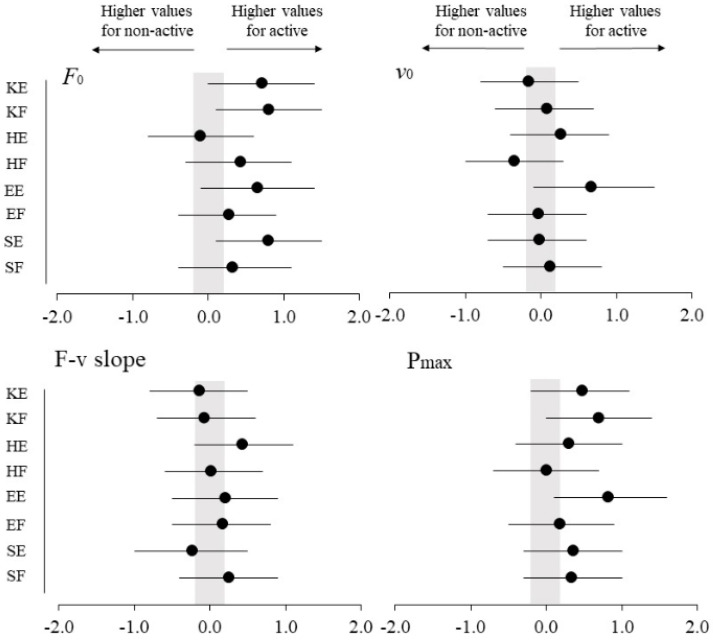
Standardized differences (Effect sizes and 95% confidence intervals) for maximum force (*F*_0_; **upper-left panel**), maximum velocity (*v*_0_; **upper-right panel**), force-velocity slope (F-v slope; **lower-left panel**) and maximum power (P_max_; **lower-right panel**) between active and non-active groups for each muscle group. ES, effect size. KE, knee extension; KF, knee flexion; HE, hip extension; HF, hip flexion; EE, elbow extension; EF, elbow flexion; SE, shoulder extension; SF, shoulder flexion. (Effect size = Active mean − Non-active mean/SDboth). The probability that the true difference was trivial (ES from −0.20 to 0.20) or substantial is depicted.

**Table 1 ijerph-17-08570-t001:** Comparison of the force-velocity relationship parameters between muscle groups and physical activity levels for each joint.

F-v Parameter	Joint	Active		Non-Active		ANOVA		
		Flexor	Extensor	Flexor	Extensor	Muscle	PAL	Muscle × PAL
*F*_0_ (N·kg^−2/3^)	Knee	20.5 ± 4.0	29.7 ± 5.0 *	17.6 ± 3.2 ^#^	26.1 ± 4.8 ^#^	*p* < 0.001	*p* = 0.015	*p* = 0.706
	Hip	19.4 ± 4.2	37.4 ± 9.8 *	19.8 ± 3.1	33.5 ± 8.0	*p* < 0.001	*p* = 0.403	*p* = 0.093
	Elbow	12.3 ± 2.0	12.1 ± 2.4	11.6 ± 2.9	10.7 ± 1.7	*p* = 0.192	*p* = 0.127	*p* = 0.387
	Shoulder	18.7 ± 3.5	14.0 ± 2.8 *	17.6 ± 2.8	12.0 ± 2.1	*p* < 0.001	*p* = 0.093	*p* = 0.372
*v*_0_ (m·s^−1^)	Knee	3.20 ± 0.93	2.50 ± 0.51 *	3.10 ± 1.59	2.65 ± 1.46	*p* = 0.009	*p* = 0.940	*p* = 0.538
	Hip	4.17 ± 2.40	3.07 ± 2.03	3.45 ± 2.75	3.93 ± 2.78	*p* = 0.556	*p* = 0.907	*p* = 0.142
	Elbow	4.39 ± 1.85	4.76 ± 2.36	4.47 ± 2.25	3.48 ± 1.36	*p* = 0.508	*p* = 0.241	*p* = 0.158
	Shoulder	3.72 ± 1.78	3.35 ± 1.49	3.42 ± 2.50	3.37 ± 2.19	*p* = 0.523	*p* = 0.804	*p* = 0.629
F-v slope(N·m·^−1^·kg^−2/3^)	Knee	6.17 ± 2.54	10.64 ± 2.82 *	5.98 ± 2.75	10.16 ± 4.30	*p* < 0.001	*p* = 0.709	*p* = 0.773
	Hip	5.50 ± 3.40	11.19 ± 5.96 *	7.01 ± 3.34	11.34 ± 6.52	*p* < 0.001	*p* = 0.543	*p* = 0.487
	Elbow	2.98 ± 1.11	2.94 ± 1.31	3.28 ± 2.53	3.22 ± 1.21	*p* = 0.884	*p* = 0.395	*p* = 0.972
	Shoulder	5.93 ± 3.07	4.58 ± 2.33 *	6.88 ± 3.98	4.11 ± 1.67	*p* < 0.001	*p* = 0.770	*p* = 0.168
P_max_ (W·kg^−2/3^)	Knee	16.1 ± 4.4	18.4 ± 3.8 *	12.9 ± 4.8 ^#^	16.2 ± 5.2	*p* = 0.001	*p* = 0.040	*p* = 0.507
	Hip	19.6 ± 10.8	29.5 ± 19.0 *	16.1 ± 11.1	29.3 ± 14.8	*p* = 0.001	*p* = 0.622	*p* = 0.622
	Elbow	13.4 ± 5.6	14.3 ± 7.5	12.2 ± 5.9	9.4 ± 4.3 ^#^	*p* = 0.459	*p* = 0.076	*p* = 0.147
	Shoulder	16.9 ± 7.7	11.3 ± 4.6 *	14.1 ± 8.5	9.6 ± 4.7 ^#^	*p* < 0.001	*p* = 0.244	*p* = 0.623

Mean ± standard deviation. *F*_0_, maximum force; *v*_0_, maximum velocity; F-v slope, force-velocity slope; P_max_, maximum power; PAL, physical activity level. *, significant differences respect to flexor; ^#^, significant differences respect to Active.

**Table 2 ijerph-17-08570-t002:** Association of the force-velocity relationship parameters between flexor and extensor muscles acting on the same joint.

	*F* _0_	*v* _0_	F-v Slope	P_max_
Knee	0.571 **	0.349 **	0.554 **	0.496 **
Hip	0.640 **	0.132	0.385 *	0.142
Elbow	0.513 **	0.057	−0.110	0.275
Shoulder	0.560 **	0.467 **	0.474 **	0.579 **

*F*_0_, maximum force; *v*_0_, maximum velocity; F-v slope, force-velocity slope; P_max_, maximum power. Statistical significance: * *p* < 0.05, ** *p* < 0.01.

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
