# Peer review of "Isokinetic Testing: Sensitivity of the Force-Velocity Relationship Assessed through the Two-Point Method to Discriminate between Muscle Groups and Participants’ Physical Activity Levels"

_ijerph, 2020, doi:10.3390/ijerph17228570_

Round 1

Reviewer 1 Report

The topic is recent and interesting, howeverthe study background, aim and results should be improved and written with greater clarity.  The practical application of the study must also be clarified. Moreover, the excessive number of abbreviations makes it difficult to read and understand the article.

Line 35 – There are other better references to this statement.

Line 36 – What do you mean with “capacities”?

Line 37 – Isokinetic strength balance ratios are very common measurements.

Line 40 – concentric and eccentric actions?

Line 41 – Which joint test?

Line 45 – This sentence needs to be improved. Isokinetic testing aim is not no test the velocity.

Line 51 – The linear relationship between Force and velocity isn’t new.

Line 54 –The relationship between power and velocity is not linear. Verify.

Line 64 – Janicijevic et al. studied it, didn’t they?

Line 66 – The paragraph needs to be rewritten in order to clarify the problem that should be solved.

Line 78 – In the first paragraph the authors presented previous studies with isokinetic strength results.

Introduction – The background can be improved.

Line 96 – The aim of the study was not clear.

Line 99 – The hypothesis is different from the aim of the study.

Line 110 – The non active group presented higher body mass. It is important to know which kind of physical activity the active group do.

Line 119 – It is important to know the study design.

Line  122 - Are the subjects familiarized with isokinetic testing?

Line 131 – Stretching exercises can be harmful for strength production, but  warm-up exercises performed together with SS before activity can avoid detrimental effects on muscle strength.  Please, consult Mascarin et al, 2015. (Stretch-Induced Reductions in Throwing Performance Are Attenuated by Warm-up Before Exercise. Mascarin NC et al. J Strength Cond Res. 2015 May;29(5):1393-8. doi: 10.1519/JSC.0000000000000752.)

Line 135 – The test position, ROM, joint and muscles tested should be described.  

Line 156 – Were the variables distribution tested?

Line 156 – How the sample size was calculated?

Line 173 – At least one more isokinetic velocity test should be evaluated to verify if the relationship is really linear.

Figure 2 and 3  – Figures needs to be better explained.

Line 216 – The study conclusions are impaired if the power of the tests is not informed.

Line 219 – The aim was not present in the introduction.

Line 229 – It is a not new finding.

Line 236 – Data about the two groups physical activity level should be presented. It is difficult to interpret the data without this information and without the information about which kind of physical activity they are used to perform.

Line 237 – Data about the reproducibility of isokinetic hip test should be cited.

Line 238 – The are too many acronyms and abbreviations in the text, which difficult the reading.

Line 237-239 – The sports injuries are not studied in the present study, therefore they should not he discussed. Moreover, why the author to write only about hamstrings injuries? What about the other injuries associated with muscular weakness or imbalances?

Line 294 – The association between the results and the conclusion should be better explained.

Author Response

REVIEWER I

COMMENT

The topic is recent and interesting, however the study background, aim and results should be improved and written with greater clarity.  The practical application of the study must also be clarified. Moreover, the excessive number of abbreviations makes it difficult to read and understand the article.

RESPONSE

We appreciate the positive comment regarding the interest of our study. We have conducted major changes in the manuscript following the specific reviewers' suggestions. Firstly, we omitted many of the abbreviations such as ROM (Range of motion), KE (Knee extension), KF (Knee flexion), HE (Hip extension), HF (hip flexion), EE (elbow extension), EF (elbow flexion), SE (shoulder extension), SF (shoulder flexion), and we think these changes improved the readability of the manuscript. Secondly, we omitted discussion about injuries, we clarified study design, added required references, clarified requested terms (i.e., such as “capacities”), etc. Finally, we rewrote aims and some sections of the introduction and method. We believe that all these changes have significantly improved the quality of our manuscript. Thank you for the detailed feedback.

COMMENT

Line 35 – There are other better references to this statement.

RESPONSE

We agree with the reviewer. We have replaced this reference by the following ones:

  1. Osternig, L.R. Isokinetic dynamometry: Implications for muscle testing and rehabilitation. Exerc. Sport Sci. Rev. 1986, 14, 45–80, doi:10.1249/00003677-198600140-00005.
  2. Mayhew, T.P.; Rothstein, J.M.; Finucane, S.D.G.; Lamb, R.L. Performance characteristics of the Kin-Com® dynamometer. Phys. Ther. 1994, 74, 1047–1054, doi:10.1093/ptj/74.11.1047.
  3. Drouin, J.M.; Valovich-McLeod, T.C.; Shultz, S.J.; Gansneder, B.M.; Perrin, D.H. Reliability and validity of the Biodex system 3 pro isokinetic dynamometer velocity, torque and position measurements. Eur. J. Appl. Physiol. 2004, 91, 22–29, doi:10.1007/s00421-003-0933-0.

COMMENT

Line 36 – What do you mean with “capacities”?

Line 37 – Isokinetic strength balance ratios are very common measurements.

RESPONSE

Lines 35-37: We agree with the reviewer that this part needed to be clarified. We modified this sentence and now it reads: “Therefore, they have been widely used to assess the maximal capacities of individual muscle groups to produce force and power as well as the balance ratios in these maximal capacities between the individual muscles [2, 3].”

COMMENT

Line 40 – concentric and eccentric actions?

RESPONSE

Lines 39-42: Here and throughout the manuscript we are referring specifically to the concentric actions because they were the actions assessed in the current study. We have modified the sentence and now it reads: “However, angular velocities above 180 °/s have been discouraged when assessing isokinetic strength during concentric actions because the range of motion under a constant velocity is very small and this may compromise the accuracy of the measurement [8].”

COMMENT

Line 41 – Which joint test?

RESPONSE

These are the two velocity most commonly used for a variety of joint test and this has been indicated in the manuscript. We provide several citations showing that the velocities have been used in different joint tests. We believe that it is not necessary to write the name of each single joint test in which these velocities have been used and the interested reader can take a look to the suggested references.

COMMENT

Line 45 – This sentence needs to be improved. Isokinetic testing aim is not no test the velocity.

RESPONSE

Lines 44-46: We have made required modifications, to make the sentence more straightforward and clearer: “However, it is known that higher force outputs can be achieved under isokinetic tests performed at lower angular velocities, while maximal power could be attained under isokinetic test performed at higher angular velocities (i.e., > 180 °/s) [11].”

COMMENT

Line 51 – The linear relationship between Force and velocity isn’t new.

RESPONSE

We agree with the reviewer’s comment. We have added additional references to support this statement. Because of the evident linear relationship between force and velocity demonstrated in previous studies, in the present study we applied the two-point method.

Cuk, I.; Mirkov, D.; Nedeljkovic, A.; Kukolj, M.; Ugarkovic, D.; Jaric, S. Force-velocity property of leg muscles in individuals of different level of physical fitness. Sport. Biomech. 2016, 15, 207–219, doi:10.1080/14763141.2016.1159724.

García-Ramos, A.; Jaric, S. Two-point method: a quick and fatigue-free procedure for assessment of muscle mechanical capacities and the one-repetition maximum. Strength Cond. J. 2018, 40, 54–66, doi:10.1519/SSC.0000000000000359.

Petrovic, M.R.; García-Ramos, A.; Janicijevic, D.N.; Pérez-Castilla, A.; Knezevic, O.M.; Mirkov, D.M. The Novel Single-Stroke Kayak Test: Can It Discriminate Between 200-m and Longer-Distance (500- and 1000-m) Specialists in Canoe Sprint? Int. J. Sports Physiol. Perform. 2020, 1–8, doi:10.1123/ijspp.2019-0925.

Alcazar, J.; Csapo, R.; Ara, I.; Alegre, L.M. On the shape of the force-velocity relationship in skeletal muscles: The linear, the hyperbolic, and the double-hyperbolic. Front. Physiol. 2019, 10, doi:10.3389/fphys.2019.00769.

COMMENT

Line 54 –The relationship between power and velocity is not linear. Verify.

RESPONSE

Lines 53-54: We have clarified this sentence and now it reads: “A direct consequence of the high linearity of the F-V relationship is the hyperbolic shape of the power-velocity relationship [15–19].”

COMMENT

Line 64 – Janicijevic et al. studied it, didn’t they?

RESPONSE

This is correct. We added this reference as well.

COMMENT

Line 66 – The paragraph needs to be rewritten in order to clarify the problem that should be solved.

RESPONSE

We made substantial changes in this paragraph, and we think the message that we wanted to transmit is clearer now.

COMMENT

Line 78 – In the first paragraph the authors presented previous studies with isokinetic strength results.

RESPONSE

The reviewer is right. However, the studies presented in the first paragraph did not report the outcomes related to the maximal velocity capacity (i.e., the velocity value attained when the force output is 0 N which is estimated through the modelling of the F-v relationship). Most of the studies did not report the maximal velocity capacity because they reported the outcomes of the individual velocities tested.

COMMENT

Introduction – The background can be improved.

RESPONSE

We have made substantial changes to the introduction section following the specific suggestions of the three reviewers. We believe that the introduction has been considerably improved thanks to the valuable comments provided by the reviewers.

COMMENT

Line 96 – The aim of the study was not clear.

RESPONSE

Lines 103-107: We have modified the aims to improve its readability and now it reads: “To address the aforementioned issues, the main aim of this study was to evaluate whether the magnitude of the F-V relationship parameters (i.e., F0, v0 and Pmax) obtained using the two-velocity method (i.e. force output recorded against only two angular velocities) are sensitive to discriminate between flexors and extensors acting on the same joints (knee, hip, elbow and shoulder), and between men with different levels of physical activity (active vs. non-active). The generalizability of the same F-v relationship parameters between antagonist muscle pairs was also examined.”

COMMENT

Line 99 – The hypothesis is different from the aim of the study.

RESPONSE

We believe that with the changes that we have implemented now it is clearer the link between the aims and hypotheses.

COMMENT

Line 110 – The non-active group presented higher body mass. It is important to know which kind of physical activity the active group do.

RESPONSE

Lines 123-125: The following sentence was added to the Participant section: “Participants were considered physically active if they were performing at least 5 hours of moderate to highly intensive physical activity per week, while non-active participants reported complete absence of physical exercise.” Please, note that participants in the active group were heterogeneous regarding the type of physical activity performed and, therefore, it is complicated to summarize the type of activity/sport they were practising.

COMMENT

Line 119 – It is important to know the study design.

RESPONSE

Line 133: We specified at the beginning of the Study design section in the revised version of the manuscript that: “A crossover study design was used to explore the feasibility…” Thank you for the suggestion.

COMMENT

Line 122 - Are the subjects familiarized with isokinetic testing?

RESPONSE

Lines 150-151: We appreciate this comment. Information regarding the familiarization of the participants with isokinetic testing was added to the revised version of the manuscript: “In total, participants performed three trials of three cycles under each testing velocity, where the first trial was used for familiarization, and other two trials were used for further statistical analysis.”

COMMENT

Line 131 – Stretching exercises can be harmful for strength production, but warm-up exercises performed together with SS before activity can avoid detrimental effects on muscle strength.  Please, consult Mascarin et al, 2015. (Stretch-Induced Reductions in Throwing Performance Are Attenuated by Warm-up Before Exercise. Mascarin NC et al. J Strength Cond Res. 2015 May;29(5):1393-8. doi: 10.1519/JSC.0000000000000752.)

RESPONSE

Thank you for this comment. This reference has been added to the revised version of the manuscript.

COMMENT

Line 135 – The test position, ROM, joint and muscles tested should be described. 

RESPONSE

Lines 156-159: The details regarding ROM, joint and muscles tested are described later in the text as follows: “The range of motion was 80º for the knee tasks (from 90° to 170°) [9], 50º for the hip tasks (from 90° to 140°) [36], 65º for the elbow tasks (from 45° to 110°) [37], and 80º for the shoulder tasks (from 90° to 170°) [36].”

COMMENT

Line 156 – Were the variables distribution tested?

RESPONSE

Lines 181-182: Yes, all variables presented a normal distribution. This result is specified in the revised version of the manuscript: “The normal distribution of the data (Shapiro-Wilk test) and the homogeneity of variances (Levene's test) were confirmed (p>0.05). “

COMMENT

Line 156 – How the sample size was calculated?

RESPONSE

Lines 117-119: We added information regarding calculation of the required sample size: “A priori analysis (G*Power 3.1.9.4) revealed that for performing the statistical analysis with a power of 0.95 and α=0.05 the total sample size should consider 36 participants and, therefore, we conservatively recruited forty young men to participate in this study.”

COMMENT

Line 173 – At least one more isokinetic velocity test should be evaluated to verify if the relationship is really linear.

RESPONSE

We agree with the reviewer’s comment. However, many other studies [6–9] demonstrated that the force-velocity relationship of the single joint movements follows a linear shape (if maximal isometric force is excluded from the model). “The linear relationship between Force and velocity isn’t new” as the reviewer has indicated in a previous comment. Consequently, we did not aim to assess the linearity of the F-v relationship in the present study because this has been confirmed in previous studies by different research groups. Instead, we wanted to explore whether the two-velocity method performed in the field conditions (i.e., applying only two testing velocities) is a sensitive method to detect differences between active and non-active subjects as well as between different antagonistic muscle groups.

COMMENT

Figure 2 and 3  – Figures needs to be better explained.

RESPONSE

We agree with the reviewer’s comment. We added additional explanations in the figure legends.

COMMENT

Line 216 – The study conclusions are impaired if the power of the tests is not informed.

RESPONSE

An a priori power calculation was included in the methods section. The conclusions presented here are based on the null-hypothesis statistical tests as well as on the magnitude of the differences assessed by the Cohen´s d effect size.

COMMENT

Line 219 – The aim was not present in the introduction.

RESPONSE

We have rewritten the aims to clearly show that we compared the magnitude of the F-V relationship parameters (i.e., F0, v0, Pmax, and F-v slope) between flexors and extensors acting on the same joints (knee, hip, elbow and shoulder), and between men with different levels of physical activity (active vs. non-active).

COMMENT

Line 229 – It is a not new finding.

RESPONSE

Yes, we agree with the reviewer. However, we think it is good to emphasize at this point that the two-velocity method was able to discriminate between magnitudes of the mechanical capacities of the knee flexors and extensors muscles.

COMMENT

Line 236 – Data about the two groups physical activity level should be presented. It is difficult to interpret the data without this information and without the information about which kind of physical activity they are used to perform.

RESPONSE

Lines 123-125: We agree that this information is very important. We have specified the differences between the active and non-active groups: “Participants were considered physically active if they were performing at least 5 hours of moderate to highly intensive physical activity per week, while non-active participants reported complete absence of physical exercise”.

COMMENT

Line 237 – Data about the reproducibility of isokinetic hip test should be cited.

RESPONSE

We have added a reference about reliability of the hip isokinetic test in the manuscript:  

Meyer, C.; Corten, K.; Wesseling, M.; Peers, K.; Simon, J.P.; Jonkers, I.; Desloovere, K. Test-retest reliability of innovated strength tests for hip muscles. PLoS One 2013, 8, 81149, doi:10.1371/journal.pone.0081149.

COMMENT

Line 238 – The are too many acronyms and abbreviations in the text, which difficult the reading.

RESPONSE

We agree that we used many abbreviations in the text. Therefore, we omitted many of them such as ROM (Range of motion), KE (Knee extension), KF (Knee flexion), HE (Hip extension), HF (hip flexion), EE (elbow extension), EF (elbow flexion), SE (shoulder extension), SF (shoulder flexion). We think that the readability of the manuscript is significantly improved now.

COMMENT

Line 237-239 – The sports injuries are not studied in the present study, therefore they should not he discussed. Moreover, why the author to write only about hamstrings injuries? What about the other injuries associated with muscular weakness or imbalances?

RESPONSE

We agree with this comment. We deleted the sentence about hamstring injuries and also the sentence regarding elbow injuries mentioned in the next paragraph.

COMMENT

Line 294 – The association between the results and the conclusion should be better explained.

RESPONSE

We presented a brief conclusion based on the results discussed above for each of our aims: (I) comparison of F-V paraments between antagonist muscle groups, (II) comparison of F-V paraments between males with different levels of physical activity, and (III) correlations between the F-v relationship parameters of antagonist muscle groups. The implications of these results are presented in the last sentence of the conclusions section. We believe that we have summarized well our main findings, but if the reviewer has specific suggestions, we will be happy to consider them as we have done with the previous comments. Thank you for the detailed review of our manuscript.

Reviewer 2 Report

General Comment: The manuscript is well written and after some minor revisions could be suitable for publication. This study could contribute to the better understanding of the isokinetic testing procedure of the F-v relationship through the two-velocity method.

Specific minor comments:

  1. In the abstract part, the conclusion “The sensitivity of the two-velocity method for testing the maximal mechanical capacities was high for the knee, moderate for the hip and shoulder, and low for the elbow joint” is not specific, please give some substantial guidance for the improvement of physical activity level.
  2. In the introduction part, line 66-69, why “It would be important to elucidate whether the F-v relationship assessed through the two-velocity method is able to discriminate between participants of different physical activity levels (active vs. non-active) as well as between antagonistic muscle groups (e.g., knee, hip, elbow and shoulder).”?
  3. From line 88-90, why “it would be also interesting to determine the possibility of generalizing the outcomes of the F-v relationship between antagonist muscle pairs”, please give some explanation.
  4. From line 288-290, “In addition, the lack of significant differences between active and non-active males for several F-v relationship parameters and muscle groups could be the consequence of not controlling the type of sport and recreational activity performed by the subjects”, please give some references on this viewpoint.

Author Response

REVIEWER 2

General Comment:

The manuscript is well written and after some minor revisions could be suitable for publication. This study could contribute to the better understanding of the isokinetic testing procedure of the F-v relationship through the two-velocity method.

RESPONSE

The reviewer’s comments regarding the quality of writing is highly appreciated. We have considered all specific comments, and we think that the quality of the paper is increased due to the implemented changes.

Specific minor comments:

COMMENT

In the abstract part, the conclusion “The sensitivity of the two-velocity method for testing the maximal mechanical capacities was high for the knee, moderate for the hip and shoulder, and low for the elbow joint” is not specific, please give some substantial guidance for the improvement of physical activity level.

RESPONSE

We appreciate the reviewer's suggestion. However, please note that the present study was not designed to provide guidelines in order to improve physical activity level, but to elucidate whether the two-velocity method can discriminate between F-v parameters obtained in antagonist muscle groups. Therefore, the conclusion of the abstract section is related to the main purpose of the study.

COMMENT

In the introduction part, line 66-69, why “It would be important to elucidate whether the F-v relationship assessed through the two-velocity method is able to discriminate between participants of different physical activity levels (active vs. non-active) as well as between antagonistic muscle groups (e.g., knee, hip, elbow and shoulder).”?

RESPONSE

Lines 339-340: We think that this sentence is very important since the two-velocity method can significantly shorten the testing procedure and it may reveal validly the magnitude of all maximal mechanical capacities. The answer to this question could be find in the conclusion section: “Therefore, since different muscle groups should be evaluated to obtain comprehensive information of the function of the whole neuromuscular system, the two-velocity method could be considered as a quick procedure for testing the maximal mechanical capacities to produce force, velocity, and power.”

COMMENT

From line 88-90, why “it would be also interesting to determine the possibility of generalizing the outcomes of the F-v relationship between antagonist muscle pairs”, please give some explanation.

RESPONSE

Lines 93-98: We clarified this sentence in the revised version of the manuscript: “Finally, it would be also interesting to determine the possibility of generalizing the outcomes of the F-v relationship between antagonist muscle pairs, since it could shorten the testing procedure (e.g., possible high correlations would allow the prediction of the mechanical capacities of one muscle group based on the results obtained in the antagonistic muscle group).”

COMMENT

From line 288-290, “In addition, the lack of significant differences between active and non-active males for several F-v relationship parameters and muscle groups could be the consequence of not controlling the type of sport and recreational activity performed by the subjects”, please give some references on this viewpoint.

RESPONSE

This sentence belongs to the limitation section of the manuscript, and it serves as a recommendation for future studies to consider this important aspect. In the present study, participants were considered physically active if they were performing at least 5 hours of moderate to highly intensive physical activity per week, while non-active participants reported complete absence of physical exercise. However, participants in the active group were heterogenous regarding the type of physical activity and sport practised and this could be a limitation.

Reviewer 3 Report

General comments

Thank you for this conducting this interesting study. The manuscript is well presented and written. Please consider my specific comments that I hope will improve the clarity and detail of the manuscript which is needed before consideration of publication.

Abstract

Line 22, where these assessments concentric or eccentric?

Could you perhaps replace the effect size vales with mean and standard deviation in order to see the differences between groups?

Introduction

Line 39, the argument of limited data with increased velocities is true; however, this is also dependent on the joint assessed based on its ROM. As such, with the knee, elbow and shoulder joints having adequate ROM, could additional velocities be assessed to better quantify the force-velocity relationship?

Lines 66-77, well developed argument here. Could you please perhaps provide rationale as to why comparing the force-velocity relationship between active and non-active, what implications would this have for those who do/do not exercise?

The introduction is well structured, provides some clear rationale, and identifies interesting gaps in the literature.

Methods

Line 109, did you do a power calculation, what is your population sample size based upon? Were group’s mass and height significantly different?

Lines 120-126, were these concentric or eccentric assessments? Please make this clearer throughout the manuscript. A total of four testing sessions, and four joints assessed. Did the authors not familiarise the participants? If so, how many visits?

What angle did the authors correct for gravity for each joint?

Why did the authors not assess eccentric strength?

For statistical analysis, I am assuming you that met the assumptions for your tests? Please can you be clearer with this.

Lines 164-165, ‘The association between the same F-v relationship parameters obtained from flexor and extensor muscle groups acting on the same joint was quantified through the r coefficient’, could this be reworded for clarity please.

Results

The results section is well presented, clear to follow, with good use of figures/tables that accompanies the text.

Based on lines 164-165, this would be clearer if that sentenced is revised.

Discussion

This section is well developed, explores and accounts for the present findings.

Is there a need to assess different or more velocities in order to see these differences in populations where larger physical differences exist?

It appears that you did not use eccentric actions. As such, could future work not consider eccentric actions due to their importance in exercise and activities of daily living?

Author Response

REVIEWER 3

GENERAL COMMENTS

Thank you for this conducting this interesting study. The manuscript is well presented and written. Please consider my specific comments that I hope will improve the clarity and detail of the manuscript which is needed before consideration of publication.

RESPONSE

The comments regarding the interesting nature of the study and quality of writing are highly appreciated. We considered all reviewer’s comments, and we made substantial changes to the manuscript. Specifically, we clarified throughout the manuscript that we evaluated only the concentric capacities of the muscles. In the limitation section we acknowledged that it would be interesting to explore whether the combination of two different angular velocities could increase the precision and sensitivity of the F-v relationship parameters as it has been raised as a concern. We have also clarified the methodological details about the familiarisation period, normal distribution of the data, procedure used for defining the number of participants, etc. A point by point response to each specific comment is provided bellow, and we think that all implemented changes improved substantially the quality of the manuscript.

Abstract

COMMENT

Line 22, where these assessments concentric or eccentric?

RESPONSE

Lines 21-24: Here and throughout the manuscript we are referring specifically to concentric actions. We have specified this issue in the revised version of the manuscript: “The concentric force output of the knee, hip, elbow, and shoulder flexors and extensors of 27 active and 13 non-active men was recorded at 60 and 180°/s to determine the F-v relationship parameters (maximum force [F0], maximum velocity [v0], and maximum power [Pmax]).”

COMMENT

Could you perhaps replace the effect size vales with mean and standard deviation in order to see the differences between groups?

RESPONSE

Due to the 200 words restriction in the abstract we decided to report the effect size. However, mean, and standard deviations are presented in the results section.

Introduction

COMMENT

Line 39, the argument of limited data with increased velocities is true; however, this is also dependent on the joint assessed based on its ROM. As such, with the knee, elbow and shoulder joints having adequate ROM, could additional velocities be assessed to better quantify the force-velocity relationship?

RESPONSE

Lines 323-326: This is a very pertinent comment and of course something to be study in the future. It makes total sense to us that the two optimal velocities for applying the two-point method could be exercise-specific depending on the ROM. We have acknowledged this in the limitation section: “Therefore, it would be interesting to explore which combination of two velocities increases the accuracy in the determination of the F-v relationship parameters. It is plausible that more extreme angular velocities could be recommended for task with longer range of motion, but this hypothesis should be confirmed by future studies.”

COMMENT

Lines 66-77, well developed argument here. Could you please perhaps provide rationale as to why comparing the force-velocity relationship between active and non-active, what implications would this have for those who do/do not exercise?

RESPONSE

Lines 69-81: Thank you for this comment. The main aim of this manuscript is to explore whether the two-velocity method is sensitive enough to detect differences in the mechanical outputs of physically active and non-active participants (as well as between antagonist muscle groups). However, to increase the clarity, we have rewritten this paragraph and now it reads: The F-v relationship parameters were effective to discriminate between jumping abilities of participants with different physical activity levels [16], however, to our knowledge, no previous study has used isokinetic dynamometry to explore whether the F-v relationship parameters obtained during single-joint movements differ between participants with different physical activity levels. In addition, the isokinetic dynamometry has been widely used to explore the strength balance ratios of the antagonist muscle groups, since it presents valuable additional information to the strength of individual muscles [25]. In this regard, higher forces have been reported for extensor muscles acting on the knee [26] and hip [27], while flexor muscles were stronger at shoulder [28] and elbow joints [29]. However, no previous study has used the two-velocity method to explore differences in the mechanical capacities of antagonistic muscle groups. Therefore, it could be interesting to explore if the F-v relationship modelling can help us discriminate not only between maximal force capacities (F0), but also between maximal power (Pmax) and maximal velocity capacities (v0) of antagonist muscle pairs.”

COMMENT

The introduction is well structured, provides some clear rationale, and identifies interesting gaps in the literature.

RESPONSE

The reviewer's comment is highly appreciated.

Methods

COMMENT

Line 109, did you do a power calculation, what is your population sample size based upon? Were group’s mass and height significantly different?

RESPONSE

Lines 117-119: We added information regarding the calculation of the required sample size in the revised version of the manuscript: “A priori analysis (G*Power 3.1.9.4) revealed that for performing the statistical analysis with the power 0.95 and α=0.05 total sample size should be consisted from 36 participants, and therefore, we conservatively recruited forty young men to participate in this study.” The body mass differed between groups (p<0.001), while there were no differences in body height (p=.146). However, we think that this information is not of vital importance, since the equipment was adjusted to the participants individual anthropometrical characteristics. In addition, active participants are expected to present a higher body mass due to a higher muscle mass.

COMMENT

Lines 120-126, were these concentric or eccentric assessments? Please make this clearer throughout the manuscript. A total of four testing sessions, and four joints assessed. Did the authors not familiarise the participants? If so, how many visits?

RESPONSE

We clarified throughout the manuscript that we performed only concentric assessments. For example in the abstract we modified the method part in the following manner (Lines 21-24): “The concentric force output of the flexor and extensor muscles acting on the knee, hip, elbow, and shoulder of 27 active and 13 non-active men was recorded at 60 and 180°/s to determine the F-v relationship parameters (maximum force [F0], maximum velocity [v0], and maximum power [Pmax]).“ We also made changes in the introduction and method sections to clarify this aspect. Participants performed only one familiarization trial at the beginning of the test, and the information regarding the familiarization period is now specified in the revised version of the manuscript (Lines 150-151): “In total, participants performed three trials of three cycles under each testing velocity, where the first trial was used for familiarization, and other two trials were used for statistical analyses.” The number of visits is specified in the Participant section: “The study consisted of four testing sessions separated by 48-72 hours. The flexor and extensor muscle groups of one joint were tested in each session against two angular velocities.”

COMMENT

What angle did the authors correct for gravity for each joint?

RESPONSE

Lines 175-178: We added information regarding gravity correction in the revised version of the manuscript: “Gravity correction was performed by placing the lever arm as close as possible to horizontal position, but avoiding putting muscle antagonistic pairs in stretched positions. Therefore, gravity correction angle was 170° for knee tasks, 140° for hip tasks, and 90° for the elbow and shoulder tasks [41].”

COMMENT

Why did the authors not assess eccentric strength?

RESPONSE

We share the reviewer’s opinion regarding the importance of the eccentric strength evaluation. However, the F-v relationship has been proven to be linear only when we consider the concentric part of the F-v relationship. Therefore, this method (i.e., two-velocity method) may not be useful for evaluating mechanical outputs during eccentric contractions.

COMMENT

For statistical analysis, I am assuming you that met the assumptions for your tests? Please can you be clearer with this.

RESPONSE

Lines 181-182: Yes, all variables presented a normal distribution, and that is specified in the revised version of the manuscript: “The normal distribution of the data (Shapiro-Wilk test) and the homogeneity of variances (Levene's test) were confirmed (p>0.05). “

COMMENT

Lines 164-165, ‘The association between the same F-v relationship parameters obtained from flexor and extensor muscle groups acting on the same joint was quantified through the r coefficient’, could this be reworded for clarity please.

RESPONSE

Lines 190-192: We agree with the reviewer. This sentence has been revised: “The r coefficients were used to quantify the magnitude of the associations between the same F-v relationship parameters obtained in the antagonistic muscle groups acting on the same joint.”

Results

COMMENT

The results section is well presented, clear to follow, with good use of figures/tables that accompanies the text.

RESPONSE

We appreciate this comment.

COMMENT

Based on lines 164-165, this would be clearer if that sentenced is revised.

RESPONSE

Lines 190-192: We agree with the reviewer. This sentence has been revised and now it reads: “The r coefficients were used to quantify the magnitude of the associations between the same F-v relationship parameters obtained in the antagonistic muscle groups acting on the same joint.”

Discussion

COMMENT

This section is well developed, explores and accounts for the present findings.

RESPONSE

We appreciate this comment.

COMMENT

Is there a need to assess different or more velocities in order to see these differences in populations where larger physical differences exist?

RESPONSE

Lines 323-326: We agree with the reviewer’s comment, and now we addressed this issue in the limitation section: “Therefore, it would be interesting to explore which combination of two velocities increases the accuracy in the determination of the F-v relationship parameters. It is plausible that more extreme angular velocities could be recommended for task with longer range of motion, but this hypothesis should be confirmed by future studies.”

COMMENT

It appears that you did not use eccentric actions. As such, could future work not consider eccentric actions due to their importance in exercise and activities of daily living?

RESPONSE

We agree that the eccentric actions have vital importance in everyday activities. However, first we need to assess in different tasks whether the F-v relationship remains linear when concentric and eccentric forces are modelled in the same F-v relationship.

Round 2

Reviewer 1 Report

I commend the authors for this revision, as they did a terrific job of listening to the feedback from the reviewers and adjusting the body of the paper appropriately.